Accepted at the ICLR 2024 Workshop on AI4Differential Equations In Science

# Application of Neural Ordinary Differential Equations for Tokamak Plasma Dynamics Analysis

**Zefang Liu**
Fusion Research Center
Georgia Institute of Technology
Atlanta, GA 30332, USA
`liuzefang@gatech.edu`

**Weston M. Stacey**
Fusion Research Center
Georgia Institute of Technology
Atlanta, GA 30332, USA
`weston.stacey@nre.gatech.edu`

## Abstract

In the quest for controlled thermonuclear fusion, tokamaks present complex challenges in understanding burning plasma dynamics. This study introduces a multi-region multi-timescale transport model, employing Neural Ordinary Differential Equations (Neural ODEs) to simulate the intricate energy transfer processes within tokamaks. Our methodology leverages Neural ODEs for the numerical derivation of diffusivity parameters from DIII-D tokamak experimental data, enabling the precise modeling of energy interactions between electrons and ions across various regions, including the core, edge, and scrape-off layer. These regions are conceptualized as distinct nodes, capturing the critical timescales of radiation and transport processes essential for efficient tokamak operation. Validation against DIII-D plasmas under various auxiliary heating conditions demonstrates the model's effectiveness, ultimately shedding light on ways to enhance tokamak performance with deep learning.

## 1 Introduction

Fusion burning plasmas (Green et al., 2003), pivotal in ITER's deuterium-tritium (D-T) fusion experiments, produce high-energy neutrons and fusion alpha particles. The latter, confined within the tokamak's magnetic field, sequentially transfer energy to electrons and ions, leading to sophisticated radiation and energy transport mechanisms like electron cyclotron radiation (ECR) and bremsstrahlung. To effectively manage the complex dynamics of energy transfer and mitigate potential thermal runaway instability, it is crucial to model the interactions between instantaneous radiation and gradual diffusion transport across the tokamak's multiple regions. This thorough understanding of energy dynamics is essential for controlling the plasma's behavior, emphasizing the need for models that accurately represent various processes across different timescales.

To address these complexities, our study introduces a multi-region multi-timescale transport model (Liu & Stacey, 2021; Liu, 2022) that leverages Neural Ordinary Differential Equations (Neural ODEs) (Chen et al., 2018) for enhanced burning plasma simulation fidelity in tokamaks. Inspired by previous researches (Hill, 2019; Stacey, 2021), our model distinguishes between the core, edge, and scrape-off layer (SOL) regions, where each region is represented as a unique node with specific energy dynamics influenced by radiation and transport processes. By employing Neural ODEs, we dynamically optimize diffusivity parameters from DIII-D tokamak experimental data, enabling precise modeling of energy interactions. This method not only facilitates a deeper understanding of the plasma dynamics but also improves model accuracy and predictive capability, particularly in capturing the intricate interactions between regions. Our application of Neural ODEs exemplifies the potential of deep learning techniques in advancing tokamak plasma analysis, providing a robust framework for future explorations in controlled thermonuclear fusion, including ITER's D-T experiments.

## 2 TOKAMAK PLASMA DYNAMICS MODEL

This section introduces a multinodal model for tokamak plasma dynamics, starting with its geometry, then detailing particle and energy balance equations for the core, edge, and scrape-off layer (SOL) nodes, and concluding with the presentation of a parametric diffusivity model.

### 2.1 MODEL GEOMETRY

In the multinodal model, a tokamak plasma is partitioned into three distinct regions, each modeled as a separate node: the core region ($0 \leq \rho \leq \rho_{\text{core}} = 0.9$), the edge region ($\rho_{\text{core}} \leq \rho \leq \rho_{\text{edge}} = 1.0$), and the SOL region ($\rho_{\text{edge}} \leq \rho \leq \rho_{\text{sol}} = 1.1$), with the minor radius $a$ and the normalized minor radius $\rho = r/a$, as illustrated in Figure 1. This division simplifies the tokamak into a torus, whose cross section is considered circular, and delineates the core, edge, and SOL as toroidal shells, separated by torus surfaces at radii $r_{\text{core}}$, $r_{\text{edge}}$, and $r_{\text{sol}}$ corresponding to surfaces $A_{\text{core}}$, $A_{\text{edge}}$, and $A_{\text{sol}}$, respectively. Distances between these regions are defined as $\Delta r_{\text{core-edge}}$ (core to edge), $\Delta r_{\text{edge-sol}}$ (edge to SOL), and $\Delta r_{\text{sol-div}}$ (SOL node center to its outer surface).

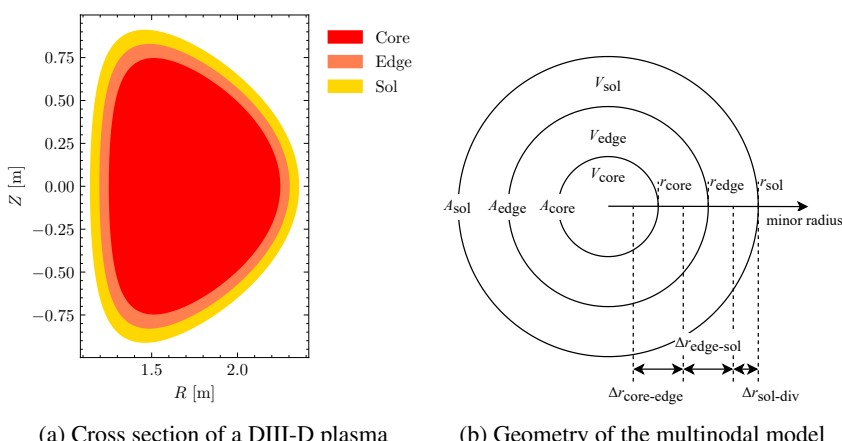

(a) Cross section of a DIII-D plasma      (b) Geometry of the multinodal model

Figure 1: Multinodal model representation of tokamak plasma regions, where the left figure shows the cross section of a DIII-D plasma, and the right figure is the simplified geometry for the multinodal model.

### 2.2 BALANCE EQUATIONS

The multinodal plasma dynamics model for DIII-D deuterium-deuterium (D-D) plasmas employs two fundamental groups of equations to capture the complex interactions within the plasma: particle and energy balance equations. Particle balance equations quantify the conservation of particles within each node accounting for particle sources and sinks, whereas energy balance equations track the thermal energy's distribution and evolution, considering heating, radiation, and energy transport. These equations form the backbone of the model, enabling detailed simulation of plasma behavior across different regions. However, this model's diffusive approach may oversimplify plasma dynamics by omitting convective (pinch) terms, with future enhancements potentially including pinch dynamics to improve accuracy, as highlighted in Angioni et al. (2009).

Particle balance equations for deuterons in the core, edge, and SOL nodes are

$$
\begin{aligned}
\frac{\mathrm{d}n_{\text{D}}^{\text{core}}}{\mathrm{d}t} &= S_{\text{D,ext}}^{\text{core}} + S_{\text{D,tran}}^{\text{core}}, \\
\frac{\mathrm{d}n_{\text{D}}^{\text{edge}}}{\mathrm{d}t} &= S_{\text{D,ext}}^{\text{edge}} + S_{\text{D,tran}}^{\text{edge}} + S_{\text{D,IOL}}^{\text{edge}}, \\
\frac{\mathrm{d}n_{\text{D}}^{\text{sol}}}{\mathrm{d}t} &= S_{\text{D,ion}}^{\text{sol}} + S_{\text{D,rec}}^{\text{sol}} + S_{\text{D,tran}}^{\text{sol}} + S_{\text{D,IOL}}^{\text{sol}},
\end{aligned}
\tag{1}
$$

where node $\in \{$ core, edge, sol $\}$, $S_{\sigma,\text{ext}}^{\text{node}}$ is the external particle source, $S_{\sigma,\text{IOL}}^{\text{node}}$ is the ion orbit loss (IOL) term (Stacey, 2011), $S_{\sigma,\text{ion}}^{\text{sol}}$ is for ionization processes, $S_{\sigma,\text{rec}}^{\text{sol}}$ is for recombination processes, and $S_{\sigma,\text{tran}}^{\text{node}}$ is the particle transport term. The particle transport for the core region is $S_{\sigma,\text{tran}}^{\text{core}} = -\frac{n_\sigma^{\text{core}}-n_\sigma^{\text{edge}}}{\tau_{P,\sigma}^{\text{core}\rightarrow\text{edge}}}$, where the internodal particle transport time is $\tau_{P,\sigma}^{\text{core}\rightarrow\text{edge}} = \frac{V_{\text{core}}\Delta r_{\text{core-edge}}}{A_{\text{core}}D_\sigma^{\text{core}}}$ with the core node volume $V_{\text{core}}$ and the core-edge interface area $A_{\text{core}}$, and $D_\sigma^{\text{core}}$ is the diffusion coefficient at $A_{\text{core}}$. Similarly, $S_{\sigma,\text{tran}}^{\text{edge}} = \frac{V_{\text{core}}}{V_{\text{edge}}}\frac{n_\sigma^{\text{core}}-n_\sigma^{\text{edge}}}{\tau_{P,\sigma}^{\text{core}\rightarrow\text{edge}}} - \frac{n_\sigma^{\text{edge}}}{\tau_{P,\sigma}^{\text{edge}\rightarrow\text{sol}}}$ and $S_{\sigma,\text{tran}}^{\text{sol}} = \frac{V_{\text{edge}}}{V_{\text{sol}}}\frac{n_\sigma^{\text{edge}}}{\tau_{P,\sigma}^{\text{edge}\rightarrow\text{sol}}} - \frac{n_\sigma^{\text{sol}}}{\tau_{P,\sigma}^{\text{sol}\rightarrow\text{div}}}$ with particle diffusivities $D_\sigma^{\text{edge}}$ and $D_\sigma^{\text{sol}}$. Other particle terms are computed by following Stacey (2012; 2021); Liu (2022). The electron densities are solved from the charge neutrality: $n_e^{\text{node}} = z_{\text{D}}n_{\text{D}}^{\text{node}} + z_z n_z^{\text{node}}$, where atomic numbers are $z_{\text{D}} = 1$ and $z_z = 6$, and the impurity density $n_z^{\text{node}}$ is obtained from experiment data.

Energy balance equations for deuterons and electrons in the core, edge, and SOL nodes are

$$
\begin{aligned}
\frac{dU_{\text{D}}^{\text{core}}}{dt} &= P_{\text{D,aux}}^{\text{core}} + Q_{\text{D}}^{\text{core}} + P_{\text{D,tran}}^{\text{core}}, & \frac{dU_e^{\text{core}}}{dt} &= P_\Omega^{\text{core}} + P_{e,\text{aux}}^{\text{core}} - P_R^{\text{core}} + Q_e^{\text{core}} + P_{e,\text{tran}}^{\text{core}}, \\
\frac{dU_{\text{D}}^{\text{edge}}}{dt} &= P_{\text{D,aux}}^{\text{edge}} + Q_{\text{D}}^{\text{edge}} + P_{\text{D,tran}}^{\text{edge}} + P_{\text{D,IOL}}^{\text{edge}}, & \frac{dU_e^{\text{edge}}}{dt} &= P_\Omega^{\text{edge}} + P_{e,\text{aux}}^{\text{edge}} - P_R^{\text{core}} + Q_e^{\text{edge}} + P_{e,\text{tran}}^{\text{edge}}, \\
\frac{dU_{\text{D}}^{\text{sol}}}{dt} &= P_{\text{D,at}}^{\text{sol}} + Q_{\text{D}}^{\text{sol}} + P_{\text{D,tran}}^{\text{sol}} + P_{\text{D,IOL}}^{\text{sol}}, & \frac{dU_e^{\text{sol}}}{dt} &= P_{e,\text{ion}}^{\text{sol}} + P_{e,\text{rec}}^{\text{sol}} - P_R^{\text{sol}} + Q_e^{\text{sol}} + P_{e,\text{tran}}^{\text{sol}},
\end{aligned}
\tag{2}
$$

where the nodal energy density is $U_\sigma^{\text{node}} = \frac{3}{2}n_\sigma^{\text{node}}T_\sigma^{\text{node}}$, $P_{\sigma,\text{aux}}^{\text{node}}$ is the auxiliary heating, $P_\Omega^{\text{node}}$ is the ohmic heating, $P_R^{\text{node}}$ is the radiative energy loss, $Q_\sigma^{\text{node}}$ is the collisional energy transfer, $P_{\sigma,\text{IOL}}^{\text{node}}$ is the IOL term (Stacey, 2011), $P_{\sigma,\text{ion}}^{\text{sol}}$, $P_{\sigma,\text{rec}}^{\text{sol}}$, and $P_{\sigma,\text{at}}^{\text{sol}}$ are for atomic and molecular processes, and $P_{\sigma,\text{tran}}^{\text{node}}$ is the energy transport term. The energy transport for the core region is $P_{\sigma,\text{tran}}^{\text{core}} = -\frac{U_\sigma^{\text{core}}-U_\sigma^{\text{edge}}}{\tau_{E,\sigma}^{\text{core}\rightarrow\text{edge}}}$, where the internodal energy transport time is $\tau_{E,\sigma}^{\text{core}\rightarrow\text{edge}} = \frac{V_{\text{core}}\Delta r_{\text{core-edge}}}{A_{\text{core}}\chi_\sigma^{\text{core}}}$ and $\chi_\sigma^{\text{core}}$ is the thermal diffusivity at the surface $A_{\text{core}}$. Similarly, $P_{\sigma,\text{tran}}^{\text{edge}} = \frac{V_{\text{core}}}{V_{\text{edge}}}\frac{U_\sigma^{\text{core}}-U_\sigma^{\text{edge}}}{\tau_{E,\sigma}^{\text{core}\rightarrow\text{edge}}} - \frac{U_\sigma^{\text{edge}}}{\tau_{E,\sigma}^{\text{edge}\rightarrow\text{sol}}}$ and $P_{\sigma,\text{tran}}^{\text{sol}} = \frac{V_{\text{edge}}}{V_{\text{sol}}}\frac{U_\sigma^{\text{edge}}}{\tau_{E,\sigma}^{\text{edge}\rightarrow\text{sol}}} - \frac{U_\sigma^{\text{sol}}}{\tau_{E,\sigma}^{\text{sol}\rightarrow\text{div}}}$ with thermal diffusivities $\chi_\sigma^{\text{edge}}$ and $\chi_\sigma^{\text{sol}}$. Similarly, other energy terms are calculated as Stacey (2012; 2021); Liu (2022).

## 2.3 Diffusivity Models

In order to calculate internodal transport times, formulas for particle and thermal diffusivities are required. One empirical scaling for the effective thermal diffusivity (Becker, 2004) in the ELMy H-mode tokamak plasma is

$$
\chi_{H98}(\rho) = \alpha_H B_T^{-3.5} n_e(\rho)^{0.9} T_e(\rho)\, |\nabla T_e(\rho)|^{1.2}\, q(\rho)^{3.0}\kappa(\rho)^{-2.9}M^{-0.6}R^{0.7}a^{-0.2}\ \left(\text{m}^2/\text{s}\right), \tag{3}
$$

where the terms in this formula are the thermal diffusivity $\chi_{H98}$ in $\text{m}^2/\text{s}$, normalized radius $\rho = r/a$, coefficient $\alpha_H = 0.123$, toroidal magnetic field $B_T$ in T, electron density $n_e$ in $10^{19}\,\text{m}^{-3}$, electron temperature $T_e$ in keV, electron temperature gradient $\nabla T_e$ in keV/m, safety factor $q = q_\psi$, local elongation $\kappa$, hydrogenic atomic mass number $M$ in $1\,\text{amu}$, major radius $R$ in m, and minor radius $a$ in m. The particle and thermal diffusivities for electrons and ions (Becker & Kardaun, 2006) are assumed as $\chi_e(\rho) = \chi_i(\rho) = \chi_{H98}(\rho)$ and $D_i(\rho) = 0.6\chi_{H98}(\rho)$, which can be replaced by separate formulas in future. This empirical scaling is used as the baseline in this study.

Besides, a parametric diffusivity formula is proposed for particle and energy diffusivities as

$$
\begin{aligned}
\frac{\chi(\rho)}{1\,\text{m}^2/\text{s}} &= \alpha_H \left(\frac{B_T}{1\,\text{T}}\right)^{\alpha_B} \left(\frac{n_e(\rho)}{10^{19}\,\text{m}^{-3}}\right)^{\alpha_n} \left(\frac{T_e(\rho)}{1\,\text{keV}}\right)^{\alpha_T} \left(\frac{|\nabla T_e(\rho)|}{1\,\text{keV/m}}\right)^{\alpha_{\nabla T}} q(\rho)^{\alpha_q}\kappa(\rho)^{\alpha_\kappa} \\
&\quad \cdot \left(\frac{M}{1\,\text{amu}}\right)^{\alpha_M} \left(\frac{R}{1\,\text{m}}\right)^{\alpha_R} \left(\frac{a}{1\,\text{m}}\right)^{\alpha_a}.
\end{aligned}
\tag{4}
$$

where $\alpha_H, \alpha_B, \dots, \alpha_a$ are diffusivity parameters for each species and node, and these parameters will be determined from experimental data. This diffusivity formula can be grouped and reformulated into a vector form: $\ln \boldsymbol{\chi}_{\text{node}} = \mathbf{b}_{\text{node}} + \mathbf{W}_{\text{node}} \ln \mathbf{x}_{\text{node}}$, where $\boldsymbol{\chi}_{\text{node}}$ is the vector of internodal

diffusivities, $\mathbf{b}_{node}$ is the bias vector, $\mathbf{W}_{node}$ is the weight matrix, and $\mathbf{x}_{node}$ is the vector of corresponding physical values.

## 3 COMPUTATIONAL FRAMEWORK

In this section, we develop a computational framework for simulating plasma dynamics in tokamaks. This framework consists of a series of interconnected modules designed to process experimental data, simulate plasma behavior, and optimize model parameters. Initially, a data module reads experimental inputs, such as two-dimensional plasma profiles and one-dimensional global parameters, from the OMFIT (Meneghini et al., 2015) for DIII-D data. A preprocessing module then standardizes these inputs into uniform time sequences and performs volume averaging on two-dimensional signals to obtain nodal particle densities and temperatures.

The core of the framework includes a diffusivity model that calculates particle and thermal diffusivities based on experimental conditions, and a transport time model that determines the timescales for particle and energy transport between nodes. These models feed into a reactor simulation module that integrates sources, sinks, and transport terms into a dynamical system, which is then solved using the Neural Ordinary Differential Equation (Neural ODE) solver (Chen et al., 2018). This dynamical system solver outputs estimated particle densities and temperatures, which are refined through an optimization module. This module computes the mean square error (MSE) between model predictions and experimental data, utilizing back-propagation for gradient computation and parameter updating via gradient descent, ensuring the model's parameters are optimized for accurate plasma behavior representation.

A workflow diagram in Figure 2 visually represents the framework's structure, where cylinders are datasets, rectangles are modules, solid lines are forward flows to solve the problem, and dashed lines are back propagation processes to optimize the parameters in the diffusivity model. This architecture enables a comprehensive analysis of tokamak plasma dynamics, leveraging modern computational techniques for enhanced modeling accuracy and predictive capability in fusion research.

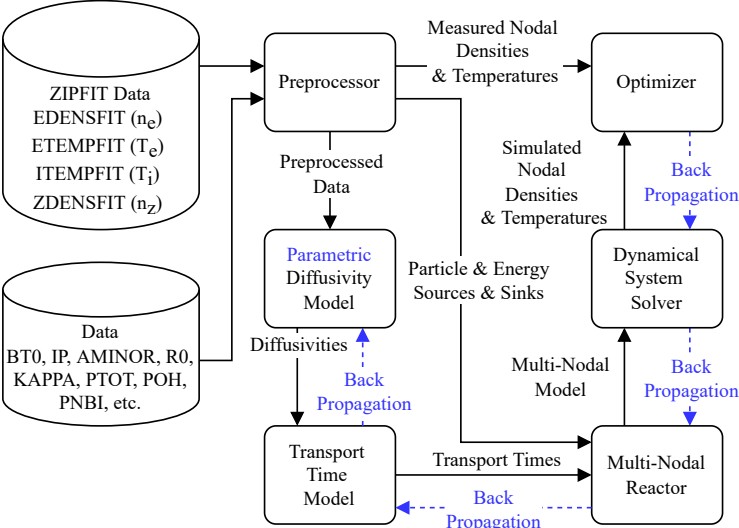

Figure 2: Computational framework diagram, including cylinders as datasets, squares as modules, solid lines as forward flows, and dashed lines as back propagation processes.

## 4 SIMULATIONS

This section details simulations of the multinodal model for DIII-D shots, covering data preparation, training and testing dataset division, model training, model evaluation, and presentation of results.

## 4.1 SIMULATION METHODS

In the development of the multinodal model for DIII-D plasma simulations, 25 shots from experimental data listed in Appendix A Table 2 are utilized, adhering to the criteria set by Hill & Stacey (2017) for ELMing non-RMP H-mode. Particle density and temperature profiles, derived from ZIP-FIT (Peng et al., 2002) integrated with EFIT and Thomson scattering data, are used to establish nodal densities and temperatures. To ensure the model's generalizability across various experimental conditions and prevent overfitting to specific shots, the data are split into two sets: a training dataset comprising 20 shots for optimizing the parametric diffusivity model, and a testing dataset of 5 shots for model performance evaluation. The training process involves solving the multinodal model's dynamical system using the training dataset, where the diffusivity parameters are initialized with the $\chi_{H98}$ model (Becker, 2004), and the nodal source and sink terms are combined into a dynamical system and solved for simulated nodal densities and temperatures. The mean squared error (MSE) between the model's predictions and experimental data is calculated to guide the optimization of diffusivity parameters through gradient descent, iterating over multiple epochs until a satisfactory MSE is achieved. Once optimized, the model's effectiveness is assessed on the testing dataset by comparing its predictions against experimental measurements, thereby evaluating the computational method's overall accuracy and reliability.

## 4.2 SIMULATION RESULTS

Following the training of the parametric diffusivity model with the designated training shots, the multinodal model's performance is evaluated on testing shots by comparing the mean squared errors (MSEs) between the model's solutions and experimental measurements, where the particle densities and temperatures are normalized by $10^{19}\,\mathrm{m}^{-3}$ and $1\,\mathrm{keV}$ respectively. The multinodal model with the empirical diffusivity scaling (Becker, 2004) is used as the baseline. This comparison, detailed in Table 1, highlights the improvements achieved through optimization, with MSE reductions exceeding 96% across individual shots and an overall average MSE decrease of more than 98%. Such significant reductions in MSEs, achieved without the model being directly trained on the testing dataset, underscore the computational method's efficacy and the optimized multinodal model's capability to generalize and accurately predict the behavior of new, unseen shots. Detailed results of the model's performance for two specific testing shots are provided in Appendix B, including experiment signals and model solutions for the core, edge, and SOL nodes. The model with optimized parameters outperforms its original, unoptimized version in performance, especially in the core and edge nodes, where the original model exhibits significant deviations between simulation results and experimental measurements.

Table 1: Mean squared errors (MSEs) of testing shots for both the multinodal model with the empirical diffusivity scaling (baseline) and the multinodal model with optimized diffusivity parameters.

| Shot | Original Model MSE | Optimized Model MSE | Relative Decrease |
|---|---|---|---|
| 131190 | 11.5861 | 0.4075 | 96.48% |
| 140418 | 56.6859 | 0.3170 | 99.44% |
| 140420 | 70.3650 | 0.5876 | 99.16% |
| 140427 | 29.7967 | 0.7105 | 97.62% |
| 140535 | 88.4208 | 0.7348 | 99.17% |
| Average | 51.3709 | 0.5515 | 98.93% |

## 5 CONCLUSION

This study introduces a multinodal plasma dynamics model for tokamak analysis, segmenting the plasma into core, edge, and SOL regions as distinct nodes with tailored timescales for various phenomena. We detail computational techniques for refining diffusivity parameters within this model, demonstrating its validation and parameter optimization using DIII-D plasma data. The enhanced performance of the optimized model signifies its advancement over previous models. This work illustrates the effectiveness of employing Neural Ordinary Differential Equations (Neural ODEs)

in tokamak plasma dynamics research, highlighting an advancement in simulation methodologies. Looking forward, applying these techniques to ITER fusion plasmas is promising but challenging due to the absence of direct training sets, potentially requiring extrapolation with derived parameters to address ITER's distinct conditions.

ACKNOWLEDGMENTS

This material is based upon work supported by the U.S. Department of Energy, Office of Science, Office of Fusion Energy Sciences, using the DIII-D National Fusion Facility, a DOE Office of Science user facility, under Award(s) DE-FC02-04ER54698.

Disclaimer: This report was prepared as an account of work sponsored by an agency of the United States Government. Neither the United States Government nor any agency thereof, nor any of their employees, makes any warranty, express or implied, or assumes any legal liability or responsibility for the accuracy, completeness, or usefulness of any information, apparatus, product, or process disclosed, or represents that its use would not infringe privately owned rights. Reference herein to any specific commercial product, process, or service by trade name, trademark, manufacturer, or otherwise does not necessarily constitute or imply its endorsement, recommendation, or favoring by the United States Government or any agency thereof. The views and opinions of authors expressed herein do not necessarily state or reflect those of the United States Government or any agency thereof.

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

## A  EXPERIMENT DATA

The DIII-D shots used in this research, following Hill & Stacey (2017), are limited to the ELMing non-RMP (non-resonant magnetic perturbation) H-mode with the standard magnetic field configuration. Basic descriptions of these shots are listed in Table 2, including the ohmic heating power $P_\Omega$, electron cyclotron heating (ECH) power $P_{\text{ECH}}$, ion cyclotron heating (ICH) power $P_{\text{ICH}}$, neutral beam injection (NBI) power $P_{\text{NBI}}$, gas puffing GAS, magnetic field $B_0$, electron density $n_e$, and electron temperature $T_e$.

Table 2: Experiment shots from DIII-D used in this study, where electron densities and temperatures are volume-averaged over the whole plasma. (Shots with * are in the testing dataset.)

| Shot | $P_\Omega$/MW | $P_{\text{ECH}}$/MW | $P_{\text{ICH}}$/MW | $P_{\text{NBI}}$/MW | GAS/(Torr · L/s) | $|B_0|$ /T | $n_e/10^{19}\,\text{m}^{-3}$ | $T_e$/keV |
|---|---|---|---|---|---|---|---|---|
| 131190* | -0.18-0.57 | 0.00-2.44 | 0.00 | 2.01-9.21 | 11.29-162.33 | 1.72-1.92 | 1.25-4.65 | 0.50-2.80 |
| 131191 | -0.11-0.26 | 0.00-2.38 | 0.00 | 2.57-9.20 | 14.11-87.21 | 1.73-1.87 | 1.08-3.89 | 0.46-3.24 |
| 131195 | 0.07-0.43 | 0.00-2.23 | 0.00 | 2.61-9.65 | 7.99-76.45 | 1.77-1.86 | 1.14-3.27 | 0.89-2.29 |
| 131196 | 0.00-0.78 | 0.00-1.27 | 0.00 | 2.02-9.79 | 11.35-84.68 | 1.76-1.87 | 1.14-3.63 | 0.49-2.63 |
| 134350 | -0.22-0.82 | 0.00-3.15 | 0.00 | 2.39-9.27 | 0.00-90.14 | 1.73-1.93 | 1.11-6.60 | 0.45-2.96 |
| 135837 | -0.06-0.58 | 0.00 | 0.00 | 0.00-14.44 | 0.05-46.83 | 1.73-2.04 | 1.22-4.85 | 0.35-1.65 |
| 135843 | 0.16-1.43 | 0.00 | 0.00 | 0.06-7.12 | 0.05-115.68 | 1.82-2.13 | 0.65-6.55 | 0.29-1.80 |
| 140417 | 0.02-0.89 | 0.00 | 0.00 | 0.61-4.37 | 0.00-70.65 | 1.90-2.02 | 1.82-5.18 | 0.48-1.44 |
| 140418* | -0.07-0.85 | 0.00 | 0.00 | 0.61-4.13 | 0.00-64.55 | 1.87-2.05 | 1.70-5.00 | 0.43-1.30 |
| 140419 | -0.12-0.82 | 0.00 | 0.00 | 0.61-4.12 | 0.00-39.96 | 1.92-2.05 | 1.43-5.26 | 0.48-1.49 |
| 140420* | 0.10-0.98 | 0.00-3.34 | 0.00 | 0.61-4.12 | 0.00-21.69 | 1.88-2.07 | 0.96-6.94 | 0.45-1.76 |
| 140421 | -0.23-0.84 | 0.00-3.23 | 0.00 | 0.61-4.11 | 0.00-17.94 | 1.90-2.06 | 0.99-5.77 | 0.57-1.76 |
| 140422 | -0.20-0.89 | 0.00 | 0.00 | 0.61-4.13 | 0.00-23.60 | 1.92-2.06 | 0.93-4.85 | 0.51-1.78 |
| 140423 | -0.13-0.86 | 0.00 | 0.00 | 0.61-4.11 | 0.00-30.99 | 1.93-2.06 | 0.96-4.44 | 0.50-1.88 |
| 140424 | -0.02-0.85 | 0.00 | 0.00 | 0.61-4.14 | 0.05-104.98 | 1.92-2.05 | 1.22-6.94 | 0.51-1.66 |
| 140425 | 0.05-0.84 | 0.00 | 0.00 | 0.61-4.14 | 0.02-112.07 | 1.90-2.06 | 1.19-7.73 | 0.42-1.49 |
| 140427* | 0.09-0.96 | 0.00 | 0.00 | 0.61-4.15 | 0.02-109.97 | 1.93-2.05 | 1.94-7.15 | 0.42-1.19 |
| 140428 | 0.10-0.87 | 0.00 | 0.00 | 0.61-4.15 | 0.00-0.20 | 1.91-2.06 | 1.50-7.06 | 0.48-1.13 |
| 140429 | 0.16-0.84 | 0.00 | 0.00 | 0.61-4.15 | 0.00-6.46 | 1.92-2.05 | 1.09-6.82 | 0.46-1.24 |
| 140430 | 0.00-0.88 | 0.00 | 0.00-0.04 | 0.61-4.15 | 0.00-26.28 | 1.93-2.08 | 1.08-4.24 | 0.48-1.61 |
| 140431 | -0.06-0.95 | 0.00 | 0.00-0.04 | 0.61-4.16 | 0.00-38.34 | 1.91-2.09 | 1.00-4.17 | 0.53-1.86 |
| 140432 | -0.08-0.95 | 0.00 | 0.00 | 0.61-4.13 | 0.00-50.36 | 1.90-2.09 | 0.98-3.85 | 0.48-2.06 |
| 140440 | 0.12-1.03 | 0.00 | 0.00 | 0.61-4.14 | 0.03-108.87 | 1.95-2.09 | 1.47-7.43 | 0.49-1.29 |
| 140535* | 0.18-0.69 | 0.00 | 0.00-0.03 | 0.63-4.48 | 0.07-118.15 | 1.92-2.10 | 1.13-2.11 | 0.23-1.29 |
| 140673 | 0.00-0.49 | 0.00-3.38 | 0.00-0.29 | 1.95-11.26 | 0.00-102.17 | 1.65-1.78 | 1.16-4.76 | 0.34-1.39 |

## B  MORE SIMULATION RESULTS

Two shots are selected from the testing set. Their experiment signals are shown in Figures 3 and 7, where several important signals, including the plasma current $I_P$, toroidal magnetic field $B_0$, safety factor $q_{95}$, gas puffing rate GAS, ohmic heating power $P_\Omega$, neutral beam injection (NBI) power $P_{\text{NBI}}$, electron cyclotron heating (ECH) power $P_{\text{ECH}}$, and ion cyclotron heating (ICH) power $P_{\text{ICH}}$, are presented. Then, their results are shown in Figures 4-6 and 8-10. In each figure, the first column is the solution from the multinodal model with the empirical diffusivity scaling (Becker, 2004), and the second column is from the multinodal model with the optimized diffusivity parameters. The densities are presented in the top row, while the temperatures are in the bottom row. The $\hat{n}_\sigma^{\text{node}}$ and $\hat{T}_\sigma^{\text{node}}$ are from the simulations of the multinodal model, and $n_\sigma^{\text{node}}$ and $T_\sigma^{\text{node}}$ are from the experiment measurements. The optimized multinodal model outperforms its original version, especially in the core and edge nodes, and more accurately tracks density and temperature with a single power source. However, improvements are needed in handling multiple power sources, such as NBI versus ICH and ECH, and in refining gas puffing distributions and addressing edge effects like ion orbit loss (IOL) (Stacey, 2011) for enhanced edge transport accuracy.

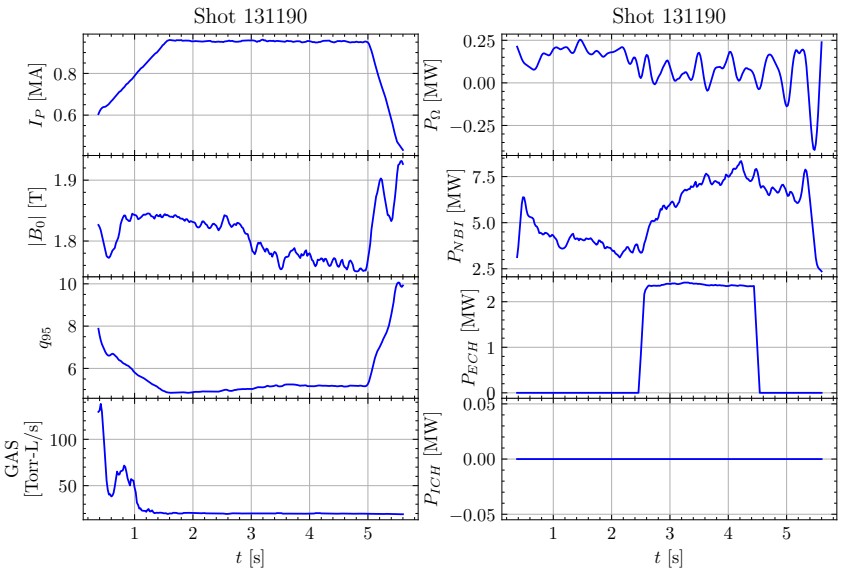

Figure 3: Signals of shot 131190, including (from top to bottom, left to right) plasma current $I_P$, toroidal magnetic field $B_0$, safety factor $q_{95}$, gas puffing rate GAS, ohmic heating power $P_\Omega$, neutral beam injection (NBI) power $P_{\text{NBI}}$, electron cyclotron heating (ECH) power $P_{\text{ECH}}$, and ion cyclotron heating (ICH) power $P_{\text{ICH}}$.

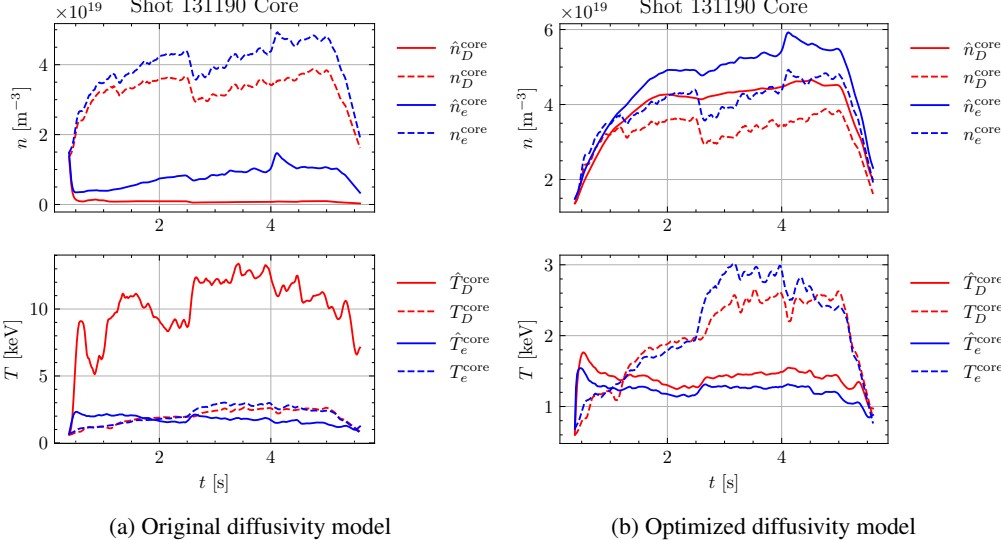

(a) Original diffusivity model      (b) Optimized diffusivity model

Figure 4: Simulation results of shot 131190 core node from both the original and optimized diffusivity models, where $\hat{n}_\sigma^{\text{node}}$ and $\hat{T}_\sigma^{\text{node}}$ are from the simulations of the multinodal model, and $n_\sigma^{\text{node}}$ and $T_\sigma^{\text{node}}$ are from the experiment measurements.

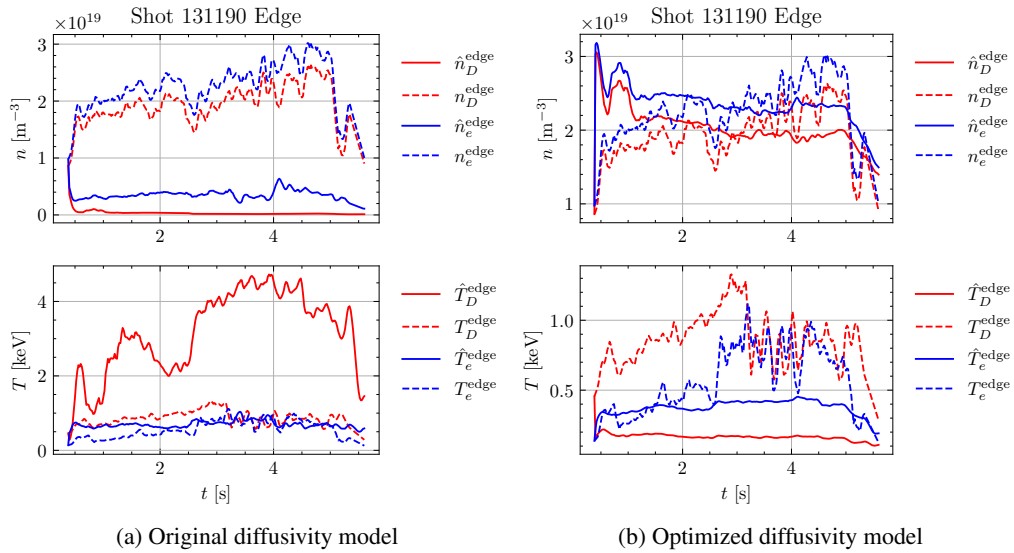

(a) Original diffusivity model

(b) Optimized diffusivity model

Figure 5: Simulation results of shot 131190 edge node from both the original and optimized diffusivity models, where $\hat{n}_\sigma^{\text{node}}$ and $\hat{T}_\sigma^{\text{node}}$ are from the simulations of the multinodal model, and $n_\sigma^{\text{node}}$ and $T_\sigma^{\text{node}}$ are from the experiment measurements.

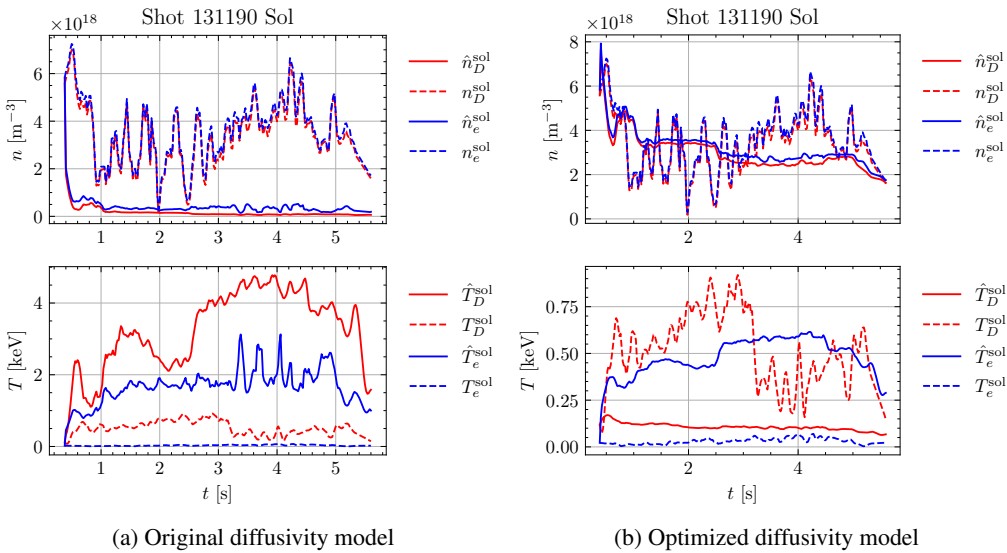

(a) Original diffusivity model

(b) Optimized diffusivity model

Figure 6: Simulation results of shot 131190 scrape-off layer (SOL) node from both the original and optimized diffusivity models, where $\hat{n}_\sigma^{\text{node}}$ and $\hat{T}_\sigma^{\text{node}}$ are from the simulations of the multinodal model, and $n_\sigma^{\text{node}}$ and $T_\sigma^{\text{node}}$ are from the experiment measurements.

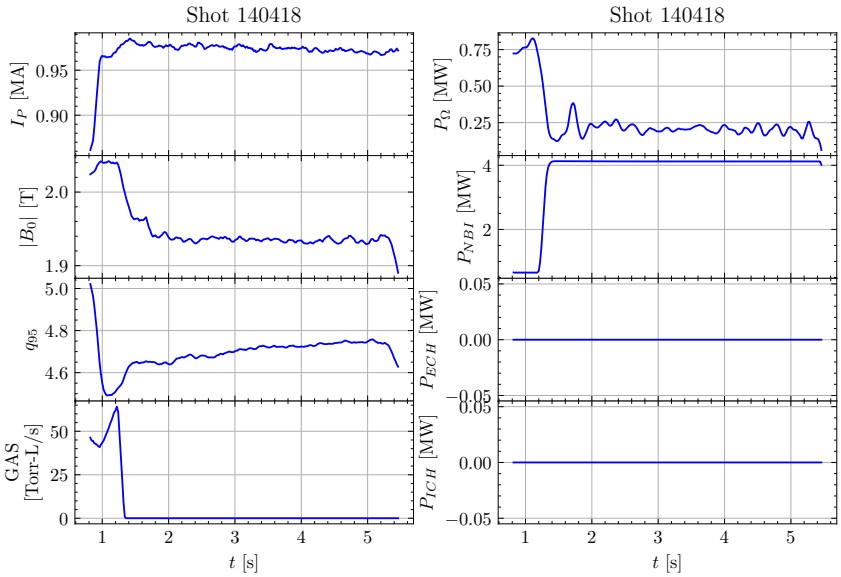

Figure 7: Signals of shot 140418, including (from top to bottom, left to right) plasma current $I_P$, toroidal magnetic field $B_0$, safety factor $q_{95}$, gas puffing rate GAS, ohmic heating power $P_\Omega$, neutral beam injection (NBI) power $P_{\text{NBI}}$, electron cyclotron heating (ECH) power $P_{\text{ECH}}$, and ion cyclotron heating (ICH) power $P_{\text{ICH}}$.

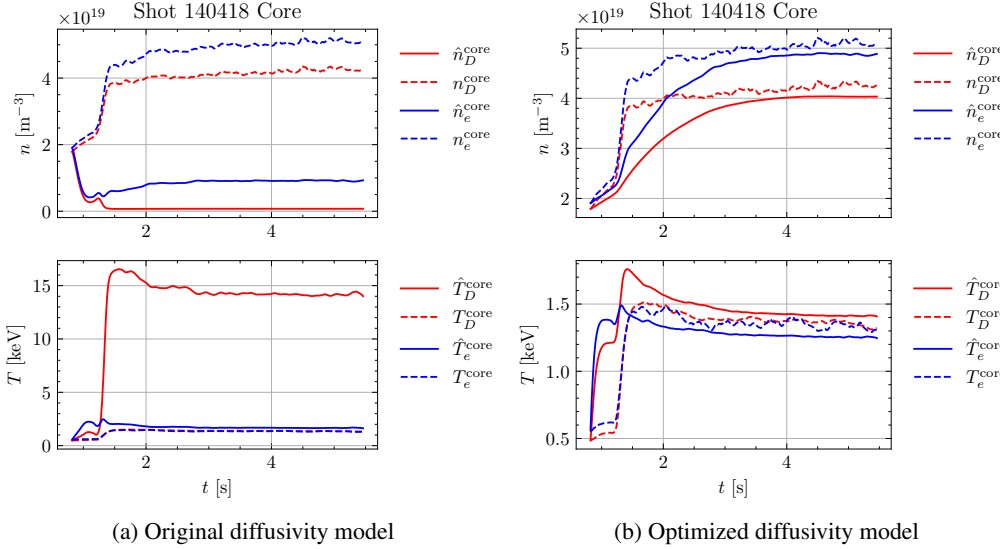

(a) Original diffusivity model

(b) Optimized diffusivity model

Figure 8: Simulation results of shot 140418 core node from both the original and optimized diffusivity models, where $\hat{n}_\sigma^{\text{node}}$ and $\hat{T}_\sigma^{\text{node}}$ are from the simulations of the multinodal model, and $n_\sigma^{\text{node}}$ and $T_\sigma^{\text{node}}$ are from the experiment measurements.

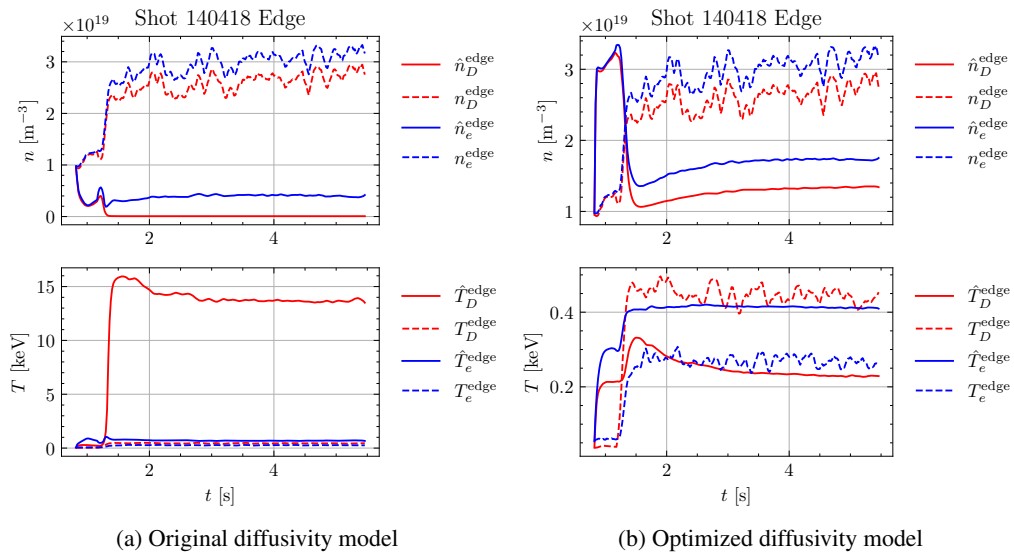

(a) Original diffusivity model

(b) Optimized diffusivity model

Figure 9: Simulation results of shot 140418 edge node from both the original and optimized diffusivity models, where $\hat{n}_\sigma^{\text{node}}$ and $\hat{T}_\sigma^{\text{node}}$ are from the simulations of the multinodal model, and $n_\sigma^{\text{node}}$ and $T_\sigma^{\text{node}}$ are from the experiment measurements.

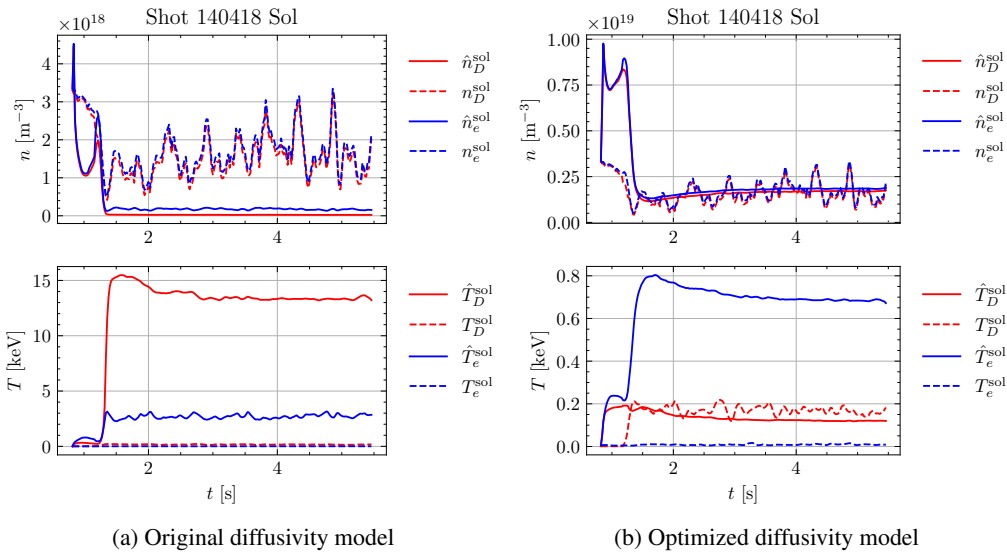

(a) Original diffusivity model

(b) Optimized diffusivity model

Figure 10: Simulation results of shot 140418 scrape-off layer (SOL) node from both the original and optimized diffusivity models, where $\hat{n}_\sigma^{\text{node}}$ and $\hat{T}_\sigma^{\text{node}}$ are from the simulations of the multinodal model, and $n_\sigma^{\text{node}}$ and $T_\sigma^{\text{node}}$ are from the experiment measurements.

