# OpenReview forum: "Application of Neural Ordinary Differential Equations for Tokamak Plasma Dynamics Analysis"
_ICLR.cc/2024/Workshop/AI4DiffEqtnsInSci — AI4DiffEqtnsInSci @ ICLR 2024 Poster_

### Official Review · Reviewer_LpgF · 2024-02-16
**Review of "Application of Neural Ordinary Differential Equations for Tokamak Plasma Dynamics Analysis"**

**Rating:** 7
**Confidence:** 3

**Review:**

This paper presents a study in which Neural ODEs are used to simulate energy transfer processes in tokamaks.  The proposed models are trained using experimental data.  Validation of the proposed data-driven models is performed under various auxiliary heating conditions.

This paper is interesting and generally well-written.  I really like that the authors are using experimental data to train/validate their models, something that is not generally done in the literature.  I do feel that there is a bit of misbalance in the paper.  The very complex equations in sections 2.2-2.3 could go in an appendix.  I would recommend more detail about the architecture and how it was optimized / trained in the body of the paper, since this is for an AI workshop. and since you are showing a tremendous improvement by optimizing your model in Table 1.  There is too much right now in the Appendix - I would suggest moving some of the figures / results here into the body of the paper, since you should really be focusing on the AI/ML.

I cannot comment on the novelty of the proposed methods for the tokamak plasma application, as this is not my area of expertise.

---

### Official Review · Reviewer_7fbj · 2024-03-01
**Better baselines are needed to accurately judge the proposed dynamics model's performance**

**Rating:** 5
**Confidence:** 2

**Review:**

Paper proposes a plasma dynamics model with a parametric module to estimate particle and energy diffusivities. The dynamical system is solved with NeuralODE and the parameters are optimized through standard backpropagation through the solver.

Since it is not my expertise, I am unable to judge the novelty and correctness of the proposed dynamics model in Section 2.
Feedback on the simulation results in Section 4:
1. Predictions for particle density $n$ seem quite far from the ground truth in the tested examples (e.g., in Figures 4, 9). But there is a mismatch with the reported MSE values in Table 1: they seem too low when compared with these predictions and the fact that $n$ is in the range of $10^{19}$.
2. The baseline (original model) seems to be the one with randomly initialized parameters. If that is indeed the case, it is a weak baseline. A better comparison would be to use other dynamical models proposed in the literature or purely neural network based methods. Without an appropriate baseline, it is hard to judge whether the reported predictions/errors are reasonable for this task.

---

### Meta-Review · Area_Chair_PEaC · 2024-03-01

**Recommendation:** Accept (Poster)

**Metareview:**

Thanks to reviewers for their honest feedback. Reviewer LpgF  made great suggestions on re-organizing the paper contents and address more useful information in the main text while moving detailed info to appendix. The authors are required to address their questions/comments in the final version. The study seems to be interesting, in terms of using Neural ODEs to simulate Tokamak plasma dynamics. Under the condition that authors will re0structure their paper for better readability, we decide to accept this as poster.

---

### Decision · Program_Chairs · 2024-03-02

Accept (Poster)